# Predictive Value of ^18^F-FDG PET/CT Using Machine Learning for Pathological Response to Neoadjuvant Concurrent Chemoradiotherapy in Patients with Stage III Non-Small Cell Lung Cancer

**DOI:** 10.3390/cancers14081987

**Published:** 2022-04-14

**Authors:** Jang Yoo, Jaeho Lee, Miju Cheon, Sang-Keun Woo, Myung-Ju Ahn, Hong Ryull Pyo, Yong Soo Choi, Joung Ho Han, Joon Young Choi

**Affiliations:** 1Department of Nuclear Medicine, Veterans Health Service Medical Center, Seoul 05368, Korea; jang8214.yoo@gmail.com (J.Y.); diva1813@naver.com (M.C.); 2Department of Preventive Medicine, Seoul National University College of Medicine, Seoul 03080, Korea; hoyajh21@gmail.com; 3Department of Nuclear Medicine, Korea Cancer Center Hospital, Korea Institute of Radiological and Medical Sciences (KIRAMS), Seoul 01812, Korea; skwoo@kirams.re.kr; 4Division of Hematology-Oncology, Department of Medicine, Samsung Medical Center, Sungkyunkwan University School of Medicine, Seoul 06351, Korea; silk.ahn@samsung.com; 5Department of Radiation Oncology, Samsung Medical Center, Sungkyunkwan University School of Medicine, Seoul 06351, Korea; hr.pyo@samsung.com; 6Department of Thoracic and Cardiovascular Surgery, Samsung Medical Center, Sungkyunkwan University School of Medicine, Seoul 06351, Korea; ysooyah.choi@samsung.com; 7Department of Pathology, Samsung Medical Center, Sungkyunkwan University School of Medicine, Seoul 06351, Korea; joungho.han@samsung.com; 8Department of Nuclear Medicine, Samsung Medical Center, Sungkyunkwan University School of Medicine, Seoul 06351, Korea

**Keywords:** non-small cell lung cancer, neoadjuvant concurrent chemoradiotherapy, ^18^F-FDG PET/CT, machine learning, random forest, pathologic complete response

## Abstract

**Simple Summary:**

The pathological complete response (pCR) after neoadjuvant chemoradiotherapy (CCRT) is an independent prognostic factor for progression-free and overall survival in non-small cell lung cancer (NSCLC). ^18^F-FDG PET/CT has been performed for initial staging work-up, treatment response, and follow-up in patients with NSCLC. Machine learning (ML) as an empirical data science has become relevant to nuclear medicine. We investigated the predictive performance of ^18^F-FDG PET/CT using an ML model to assess the treatment response to neoadjuvant CCRT in patients with stage III NSCLC, and compared the performance of the ML model predictions to predictions from conventional PET parameters and from physicians. The predictions from the ML model using radiomic features of ^18^F-FDG PET/CT provided better accuracy than predictions from conventional PET parameters and from physicians for the neoadjuvant CCRT response of stage III non-small cell lung cancer.

**Abstract:**

We investigated predictions from ^18^F-FDG PET/CT using machine learning (ML) to assess the neoadjuvant CCRT response of patients with stage III non-small cell lung cancer (NSCLC) and compared them with predictions from conventional PET parameters and from physicians. A retrospective study was conducted of 430 patients. They underwent ^18^F-FDG PET/CT before initial treatment and after neoadjuvant CCRT followed by curative surgery. We analyzed texture features from segmented tumors and reviewed the pathologic response. The ML model employed a random forest and was used to classify the binary outcome of the pathological complete response (pCR). The predictive accuracy of the ML model for the pCR was 93.4%. The accuracy of predicting pCR using the conventional PET parameters was up to 70.9%, and the accuracy of the physicians’ assessment was 80.5%. The accuracy of the prediction from the ML model was significantly higher than those derived from conventional PET parameters and provided by physicians (*p* < 0.05). The ML model is useful for predicting pCR after neoadjuvant CCRT, which showed a higher predictive accuracy than those achieved from conventional PET parameters and from physicians.

## 1. Introduction

Lung cancer is the most common malignant tumor and remains the leading cause of cancer-related death worldwide in spite of major advances in prevention and multimodal treatment [1]. Non-small cell lung cancer (NSCLC) accounts for more than 85% of all lung cancers and about 30% of NSCLC present with locally advanced disease in stage III [2]. Patients with stage III NSCLC are usually considered as inoperable. Neoadjuvant concurrent chemoradiotherapy (CCRT) followed by surgery has been established as being able to improve the overall outcome by reducing the rate of local failures and distant metastasis [3,4]. 

In patients receiving neoadjuvant CCRT for stage III NSCLC, surgical resection allows for the identification of the histopathologic tumor response to determine the prognosis and to evaluate postoperative therapeutic options. According to previous studies, the pathologic complete response (pCR) after neoadjuvant CCRT is an independent prognostic factor for progression-free and overall survival in NSCLC [5,6]. Although several papers have reported a wide range of pCR values of 16–27%, it is clear that the pCR is highly correlated with patient survival [7,8,9,10].

^18^F-fluorodeoxyglucose positron emission tomography/computed tomography (^18^F-FDG PET/CT) has been performed for initial staging work-up, treatment response, and follow-up in patients with NSCLC. It has also been viewed as appropriate for the precise investigation of treatment response after CCRT [11,12]. Previous studies have focused on the comparison of quantitative PET parameters such as the standard uptake value (SUV) after neoadjuvant treatment and histopathologic findings after surgery [13,14]. Moreover, the application of the PET response criteria in solid tumors (PERCIST 1.0) as an evaluation for ^18^F-FDG PET/CT has been performed to enhance the limitation of anatomic tumor response metrics [15,16]. The role of ^18^F-FDG PET/CT still needs to be explored because possible misinterpretations due to radiation-induced inflammation such as pneumonitis can cause problems in ^18^F-FDG PET/CT images [17,18]. 

Machine learning (ML) as an empirical data science, which can learn patterns or characteristics from one set of given data and use them to evaluate new data, has become relevant to nuclear medicine. Our previous study demonstrated that ML is well suited to performing analyses of high dimensionality radiomic feature extraction from ^18^F-FDG PET/CT, and ML analysis provided better diagnostic performance than physicians for evaluating metastatic mediastinal lymph nodes in NSCLC [19]. Although assessing the radiomic features of a tumor in clinical practice has some challenges because of the time, effort, and skill involved, we have shown that ML can improve the diagnostic accuracy and its availability in NSCLC. However, there is still no study that has evaluated the predictive performance of ML for the neoadjuvant CCRT response using the radiomic features of ^18^F-FDG PET/CT. 

Therefore, we investigated the predictive performance of ^18^F-FDG PET/CT using an ML model to assess the treatment response to neoadjuvant CCRT in patients with stage III NSCLC, and compared the performance of the ML model predictions to predictions from conventional PET parameters and from physicians. 

## 2. Materials and Methods

### 2.1. Subjects

We retrospectively reviewed the medical records of all patients newly diagnosed with stage III NSCLC through imaging studies such as chest X-ray, enhanced chest CT, and ^18^F-FDG PET/CT, as well as pathologic studies including endobronchial ultrasound-guided transbronchial needle aspiration, mediastinoscopic biopsy, or thoracotomy, between November 2008 and October 2020. To be included in the study population, patients needed to complete a planned neoadjuvant CCRT and undergo curative-intent surgical treatment for stage III NSCLC according to the 7th edition of the TNM classification [20], and undergo a second ^18^F-FDG PET/CT within approximately 3 weeks following the completion of neoadjuvant CCRT for restaging work-up. Patients in poor cardiopulmonary condition that precluded surgery or who had previously been treated because of another malignant disease were excluded from the study population. Patients who received neoadjuvant chemotherapy or radiotherapy alone were also excluded. 

This study was approved by the institutional review board of our institution (IRB No. 2020-09-185), and the requirement for informed patient consent was waived due to its retrospective design. 

### 2.2. Neoadjuvant CCRT and Histopathologic Evaluation

The neoadjuvant CCRT consisted of chemotherapy and concurrent thoracic radiotherapy. Thoracic radiotherapy was delivered to patients with a total dose of 45 Gy with 1.8 Gy/fraction over 5 weeks from November 2008 to October 2009 or 44 Gy with 2.0 Gy/fraction over 4.5 weeks using 10-MV X-rays from October 2009 and thereafter. The radiotherapy target volume included the known gross and clinical disease plus adequate peripheral margins. The chemotherapy regimens mostly consisted of intravenous administration of paclitaxel (50 mg/m^2^ per week) or docetaxel (20 mg/m^2^ per week) plus either cisplatin (25 mg/m^2^ per week) or carboplatin (AUC, 1.5/week) for 5 weeks. The first dose of chemotherapy was delivered on the first day of thoracic radiotherapy [3,4,21]. 

Surgical procedures were planned for 4~6 weeks following the completion of neoadjuvant CCRT and comprised resection of the affected lung plus mediastinal lymph nodes dissection, depending on the clinical stage. Pulmonary resection included lobectomy, bilobectomy, pneumonectomy, or lobectomy with en bloc wedge resection according to the extent of the primary tumor. After surgical resection, the specimens were examined by pathologists for residual tumors based on hematoxylin and eosin-stained slides. They reported the percentage of residual tumor, which was determined by comparing the estimated cross-sectional area of the viable tumor foci with the estimated cross-sectional areas of necrosis, fibrosis, and inflammation on each slide. The absolute viable tumor extent was also assessed based on their calculation, and pathologic complete response (pCR) was defined as no residual viable tumor remaining in the post-therapy pathology specimen [22,23]. 

### 2.3. ^18^F-FDG PET/CT Analysis

All patients fasted for at least 6 h before ^18^F-FDG PET/CT was performed to keep their blood glucose level below 200 mg/dL. Torso PET and unenhanced CT images were acquired using a dedicated PET/CT scanner (Discovery STe, GE Healthcare, Waukesha, WI, USA) approximately 60 min after intravenous injection of 5.5 MBq/kg of ^18^F-FDG. CT images were obtained using a 16-slice helical CT with the following settings: 140 keV, 30–170 mAs with Auto A mode, and a slice section of 3.75 mm. PET images were acquired from head to thigh and attenuation-corrected PET images (voxel size, 3.9 × 3.9 × 3.3 mm^3^) were reconstructed using a 3D ordered-subset expectation-maximization algorithm (20 subsets, 2 iterations). 

For quantitative analysis, the volume of interest (VOI) from the primary tumor was delineated using the gradient-based segmentation method (PET Edge) in MIM version 6.4 (MIM Software Inc., Cleveland, OH, USA) [19]. These VOIs were saved as a DICOM-RT structure that was imported into the Chang-Gung Image Texture Analysis toolbox (CGITA, http://code.google.com/p/cgita, accessed on 1 March 2020) facilitated by MATLAB software (version 2014b; MathWorks, Inc., Natick, MA, USA) to extract the radiomic features from the PET images (Appendix A) as well as conventional PET parameters, including the maximum SUV (SUVmax), mean SUV (SUVmean), metabolic tumor volume (MTV), and total lesion glycolysis (TLG). We also calculated the differences of these conventional parameters between PET1 and PET2 by subtracting PET2 parameters from those of PET1 and dividing by those of PET1. 

Two nuclear medicine physicians (J.Y.C. and B.T.K) with more than 15 years of experience in PET/CT interpretation assessed the neoadjuvant treatment response according to PERCIST 1.0 [16] by means of a baseline ^18^F-FDG PET/CT (PET1) and second PET/CT (PET2) undertaken before surgery. They categorized all patients into four response criteria: complete metabolic response (CMR), partial metabolic response (PMR), stable metabolic disease (SMD), and progressive metabolic disease (PMD). After that, the accuracy of the predicted CMR results were compared to histopathologic pCR. 

### 2.4. Machine Learning (ML) Model

The ML model was developed as a binary classification. First, data were partitioned into a training dataset (70%) for model building and an independent testing dataset (30%) for internal validation. We developed an ML tree-based boosting model for pCR prediction using a random forest (RF) algorithm, which consisted of a multitude of decision trees and used an ensemble method to decide the outcome. Our model was trained with the bagging method to predict the pCR. Different numbers of trees were used to classify the binary decision of the result to achieve the best performance score. The Gini impurity was measured to the quality of a split. The maximum depth of the tree was 5, and the square root of the number of the features was considered for the max. number of features to look for the best split of the model. We applied a random grid search method to determine the optimal hyperparameter of the RF model [24,25,26,27]. A 10-fold cross-validation in the training dataset, a technique for reducing the bias that can occur as a result of using a single training set, was applied for method validation. All ML statistical analyses were performed using Python (version 3.8.3). 

In classic oversampling techniques, the minority data are simply replicated from the minority data population. The ML model does not reflect on variation from the oversampling data. Therefore, we tried to use SMOTE (Synthetic Minority Oversampling Technique) to deal with this class problem. This technique helped with unbalanced data by creating new synthetic data to provide balance in the distribution. SMOTE starts by choosing random data from the minority class. Then it uses a K-Nearest Neighbor (KNN) algorithm to set new points of the data. Next, new synthetic data are created between the random data and new point, which is derived from KNN algorithm. This process is repeated until the minority class reaches the same size as the majority class. Therefore, we added 322 more participants from the existing raw data. A total of 752 participants were analyzed using this oversampling technique.

Several useful scaling techniques (Min–Max scaler, Normalization, Standardization) prevent overflow and underflow of the data. They help to compare dimensional data more efficiently through a scaling process. The process reduces the conditional number of covariance matrices from the independent variables. This reduction enhances the speed of conversion and stability of the model during the optimization process. We used a standard scaler, which removes the mean and helps to scale the value’s unit variance. To adjust for the different scales of the features, standardization of the variables is necessary for the preprocessing steps.

For feature selection, top 10, 20, and 30 variables among 144 variables were selected according to the importance of the variables based on the mean decrease impurity (MDI). MDI or Gini importance was calculated as the decrease in node impurity weighted by the probability of reaching the node. The sum over the number of splits decided the variable importance of the model. The higher value of MDI meant the critical feature in the model. 

### 2.5. Statistical Analysis

The association between conventional PET parameters and pCR was determined by an independent *t*-test or the Mann–Whitney test according to the Kolmogorov–Smirnov test. Receiver operating characteristic (ROC) curve analysis was performed to assess optimal cutoff values of continuous variables using the MedCalc software package (Ver. 9.5, MedCalc Software, Mariakerke, Belgium). The predictive performance of conventional PET parameters and physicians’ diagnostic results were reported using sensitivity (Sen), specificity (Spe), positive predictive value (PPV), negative predictive value (NPV), and accuracy (ACC). 

For predictive performance of the ML model, we measured the areas under curve (AUCs), ACC, F1 score, precision (also called PPV), and recall (also known as Sen). We compared the measured values with those of predictions from conventional PET parameters and from physicians by using a McNemar test or Fisher’s exact test. A *p*-value of less than 0.05 was considered statistically significant. 

## 3. Results

### 3.1. Subject Characteristics

Among 484 consecutive patients, 430 patients were enrolled in this study. Fifty-four patients were excluded from the analysis due to a lack of surgical treatment after completion of neoadjuvant CCRT (Figure 1). The clinical characteristics of the 430 patients are summarized in Table 1. The patients were predominantly male (71.9%), and there was a high prevalence (67.2%) of adenocarcinoma among the patients. After neoadjuvant CCRT followed by surgery, the mean percentage of viable tumor in the pathologic specimen was 28.8% (range 0–95%). The pCR was observed in 54 patients (12.6%). According to PERCIST criteria, 16.7% of patients had CMR (*n* = 72). 

### 3.2. Predictive Performance of ML Model for pCR

The radiomic feature importance was obtained using a Gini index representing the coefficient of the attributes on the prediction model, as listed in Figure 2. The overall prediction performance of the ML model was compared by calculating each of the PET1 and PET2 features separately, and all variables from both PET1 and PET2 (PET3) were analyzed (Table 2). The AUCs determined by the ML model were 0.934 in PET1, 0.975 in PET2, and 0.977 in PET3. For comparison ROC curve analysis (Figure 3), the AUCs of PET2 and PET3 were significantly higher than that of PET1 (*p* = 0.009, *p* = 0.006, respectively). However, there was no significant difference between the AUCs of PET2 and PET3 (*p* = 0.805). According to other indices, PET3 revealed a better predictive performance than those results with either PET1 or PET2 variables. 

Additionally, we investigated the predictive results from the ML model using four feature subsets with the top 10, 20, 30, and all features from PET3 (Appendix A and Appendix A). The ML model outperformed other methods when all features were selected (AUC = 0.977, ACC = 0.934, F1 = 0.940, Precision = 0.937, Recall = 0.944). 

### 3.3. Predictive Performances of Conventional PET Parameters and Physicians for pCR Prediction

In conventional PET parameters, the SUVmax, SUVmean, MTV, and TLG of PET1 and the SUVmax and SUVmean of PET2 were significantly associated with the pCR (*p* < 0.05). The difference between PET1 and PET2 of the SUVmax (*p* < 0.001), SUVmean (*p* < 0.001), MTV (*p* = 0.003), and TLG (*p* < 0.001) were also significantly associated with the pCR. In contrast, the MTV and TLG of PET2 were not statistically associated with the pCR (Table 3). 

The optimal cutoff values that allowed significant association with the pCR were PET1-SUVmax = 13.15, PET1-SUVmean = 4.70, PET1-MTV = 41.11, PET1-TLG = 142.97, PET2-SUVmax = 3.97, PET2-SUVmean = 1.83, dSUVmax = 56.5%, dSUVmean = 43.9%, dMTV = 55.4%, and dTLG = 86.2%. Using these cutoff values, the predictive performance of the PET parameters are listed in Table 4. The predictive performance of the physicians based on their diagnostic result are also presented in Table 4. 

### 3.4. Comparisons of the ML Model with Conventional PET Parameters and Physicians

A comparison of the predictive performances between conventional PET parameters, physicians, and the ML model are shown in Table 4. First, the performance of the ML model for pCR prediction was compared with those of conventional PET parameters by analyzing the AUCs. The ML model revealed higher AUC values than all of the single PET parameters (*p* < 0.001). When the pCR was predicted with the conventional single PET parameter, the AUC was only 0.588 to 0.745. By applying the ML model using variable radiomic features, however, the AUC improved to 0.977. In terms of predictive performance, the ML model showed significantly higher performance in Spe, PPV, and ACC than was achieved with any of the conventional PET parameters (*p* < 0.001). When comparing the predictive performances of physicians and of the ML model, the ACC of the ML model was significantly higher than that of physicians (93.4 vs. 80.5%, *p* < 0.001). Not only ACC, but also Sen, Spe, and PPV showed that the ML model significantly increased the results of physicians (94.4 vs. 33.9%, *p* < 0.001; 92.2 vs. 86.4%, *p* = 0.001; 93.7 vs. 29.2%, *p* < 0.001; respectively). NPV was the only case where there was no significant difference between the ML model and prediction by physicians (93.1 vs. 90.8%, *p* = 0.155).

## 4. Discussion

We have demonstrated that the ML model using an RF algorithm could be robust and useful in determining the pCR following neoadjuvant CCRT by radiomic features of ^18^F-FDG PET/CT. Although several studies evaluating ML for treatment response have been published recently [28,29,30,31], they mainly conducted research with multiparametric MRI features and not with ^18^F-FDG PET/CT. Only a few studies have used ^18^F-FDG PET/CT features to assess neoadjuvant treatment response in breast and rectal cancer using ML models [26,27]. To the best of our knowledge, this is the first study to predict the response to neoadjuvant CCRT in patients with NSCLC using an ML model. 

The response to neoadjuvant CCRT is critical because it affects postoperative treatment and individual prognosis. Furthermore, the correct prediction of the pCR can determine which patients will require more or less aggressive adjuvant treatment to reduce the risk of complications. Despite improvements in therapeutic modalities of neoadjuvant CCRT, the pCR rate still remains with a variety of outcomes. The gold standard for assessing the pCR is based on postoperative histopathologic findings, which could be inefficient to implement in all patients with advanced NSCLC. Therefore, it is necessary to develop a method of improving the predictive significance of non-invasive imaging modalities for establishing a personalized therapeutic strategy.

Radiomics is an emerging field where various imaging modalities are performed to extract features that may reflect changes in human tissues at the cellular levels and estimate detailed information on tumor biology and microenvironment in nuclear medicine [32,33]. The radiomic features delineated on PET/CT images can represent tumor heterogeneity including fractal dimension, tumor shape, and proliferation [34]. In our experiments, voxel statistics of radiomic features were highly ranked in the prediction for the pCR, followed by texture spectrum and co-occurrence matrix. Although there are differences in the feature importance of many radiomic variables, the ML model using them demonstrated better predictive performance for the pCR than the single conventional PET/CT parameters. Conventional PET parameters and their changes in FDG uptake before and after CCRT have been previously evaluated in determining the treatment response in patients with NSCLC [11]. We also performed these analyses; however, the ACC of the predictive performance using them was only shown to be 44.2–70.9%. Therefore, it seemed unfavorable to evaluate the predictive performance using single PET parameters even though they were statistically significantly correlated with the pCR. 

The ML model significantly outperformed the physicians in terms of Sen, Spe, PPV, and ACC. The outcomes of conducting the ML model with PET2 data revealed higher predictive performance than those of the ML model with PET1 data. It appears that radiomic features obtained from PET/CT after neoadjuvant CCRT have more relevant clinical value in the prediction of the pCR. Compared to the results of the ML model with only the variables from each time of PET/CT images, the predictive performance also increased by inputting all variables from both PET1 and PET2. We assumed that the improvement in performance is probably because of the feature importance for predicting the pCR, which is somewhat different between radiomics of PET1 and PET2. If more significant variables were input into the ML model, the predictive performance may be further improved. The PET-based radiomics can provide the potential to characterize intratumoral heterogeneity indicating resistance to neoadjuvant CCRT. Therefore, it is clinically important to evaluate treatment response not only to obtain baseline PET/CT images but also to examine PET/CT after neoadjuvant CCRT. As the current study demonstrated, the use of ML with radiomics features could be predictive of treatment response and thus help to select a more aggressive treatment for those with high-risk factors after curative surgery in patients with stage III NSCLC. 

This study had several limitations. First, this study was conducted in a retrospective manner with a limited sample size from a single center. Because radiomic features can be highly dependent on reconstruction methods and imaging parameters [35], it is planned to obtain a prospective multicenter trial to be more generalizable in the future. Second, the study population was composed of patients with different therapeutic schemes. Although we addressed a homogeneous population of patients with stage III NSCLC, it is also needed to select patients with a more uniform therapeutic modality based on the consistent guideline. Third, various pulmonary side effects can arise after radiotherapy, such as pneumonitis or fibrosis, which may challenge the response assessment, although we tried our best to exclude the possibility of treatment-induced inflammatory changes based on the relative intensity and distribution of FDG uptake in the lung parenchyma and automatically generated tumor VOI [36]. Finally, although the proposed ML model was analyzed using a 10-fold cross-validation for minimizing overfitting instead of splitting the dataset into training and test sets, external validation using an independent dataset is necessary to verify the clinical significance using a larger cohort. 

## 5. Conclusions

In conclusion, the developed ML model using an RF algorithm and ^18^F-FDG PET/CT radiomics features was useful for predicting the pCR after neoadjuvant CCRT in NSCLC. The predictions of the ML model had higher accuracy than predictions from conventional PET parameters and from physicians. The ML model using radiomics features can be used to facilitate the preoperative individualized prediction for the pCR. Our findings further highlight the potential, non-invasive, and effective clinical significance of an ML model to predict the pCR in patients with stage III NSCLC who had received neoadjuvant CCRT followed by surgery. 

## Figures and Tables

**Figure 1 cancers-14-01987-f001:**
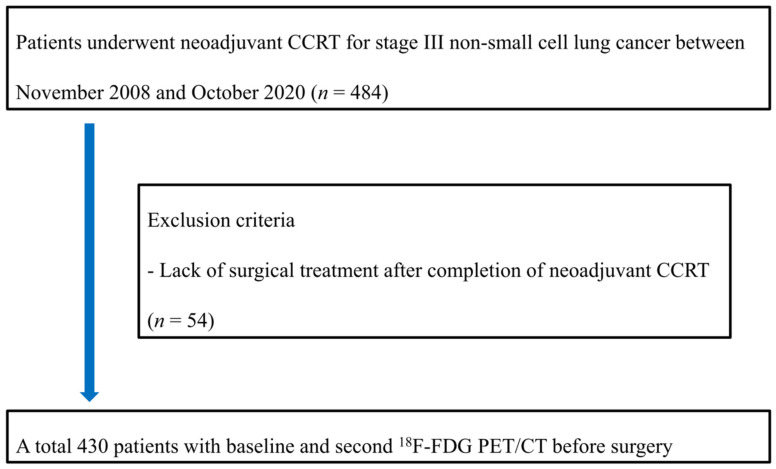
Flowchart of the inclusion and exclusion criteria for the patients.

**Figure 2 cancers-14-01987-f002:**
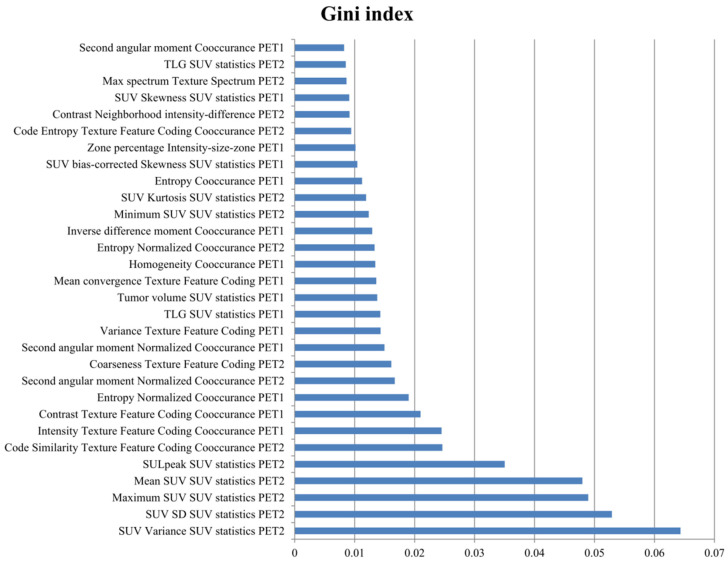
The top 30 important radiomic features from ^18^F-FDG PET/CT for pCR prediction after neoadjuvant CCRT.

**Figure 3 cancers-14-01987-f003:**
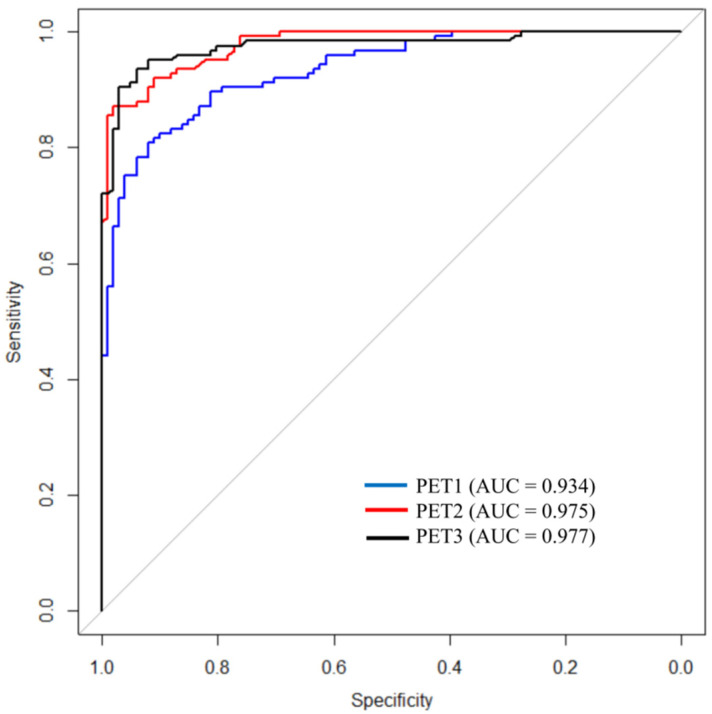
Comparisons of the ROC curves of the ML models according to the included PET data. It showed that the AUC of ML using PET/CT data obtained after neoadjuvant CCRT was significantly higher than that of using only baseline PET/CT data (*p* < 0.05).

**Table 1 cancers-14-01987-t001:** Subjects’ characteristics.

Characteristics		No.
Sex	Male	309 (71.9%)
	Female	121 (28.1%)
Age (years)	Mean (range)	61.8 (31.1–79.5)
Tumor pathology	Adenocarcinoma	289 (67.2%)
	Squamous cell carcinoma	125 (29.1%)
	Others	16 (3.7%)
Stage (AJCC 7th)	IIIa	415 (96.5%)
	IIIb	15 (3.5%)
Type of surgery	Lobectomy	339 (78.8%)
	Bilobectomy	32 (7.4%)
	Pneumonectomy	23 (5.4%)
	Lobectomy with en bloc wedge resection	36 (8.4%)
Viable tumor on pathologic specimen	Mean % (range)	28.8 (0–95.0)
Pathologic response	pCR	54 (12.6%)
	Non-pCR	376 (87.4%)
Response by PERCIST	CMR	72 (16.7%)
	PMR	281 (65.4%)
	SMD	72 (16.7%)
	PMD	5 (1.2%)

pCR, pathologic complete response; PERCIST, PET response criteria in solid tumors; CMR, complete metabolic response; PMR, partial metabolic response; SMD, stable metabolic disease; PMD, progressive metabolic disease.

**Table 2 cancers-14-01987-t002:** Comparisons in predictive performance of the ML models using a random forest algorithm for pCR prediction with the included PET data.

ML Model	AUC	ACC	F1	Precision	Recall
PET1	0.934 *^,†^	0.827 *^,†^	0.853 *^,†^	0.802 *^,†^	0.912 ^†^
PET2	0.975 *	0.902 *^,‡^	0.912 *^,‡^	0.905 *^,‡^	0.920
PET3	0.977 ^†^	0.934 ^†,‡^	0.940 ^†,‡^	0.937 ^†,‡^	0.944 ^†^

AUC, area under curve; ACC, accuracy; PET3, combining PET1 and PET2; *, †, ‡, *p* < 0.05.

**Table 3 cancers-14-01987-t003:** Comparisons in conventional PET parameters according to the presence of pCR.

			Pathologic Response		*p*-Value
			pCR	Non-pCR	
PET1	SUVmax	Median	13.59	11.58	0.029 *
		IQR	10.01–17.47	8.35–15.53	
	SUVmean	Median	5.91	5.28	0.037 *
		IQR	4.86–7.48	3.97–6.69	
	MTV (cm^3^)	Median	42.96	21.13	0.003 *
		IQR	16.02–74.89	7.38–47.48	
	TLG	Median	223.26	113.63	0.001 *
		IQR	96.29–436.26	30.77–279.36	
PET2	SUVmax	Median	3.17	4.57	<0.001 *
		IQR	2.22–4.13	2.92–6.98	
	SUVmean	Median	1.69	2.35	<0.001 *
		IQR	1.43–2.15	1.74–3.33	
	MTV (cm^3^)	Median	10.40	8.71	0.327
		IQR	3.64–27.11	3.64–19.46	
	TLG	Median	19.42	22.00	0.475
		IQR	6.32–47.35	8.61–56.52	
Delta PET parameters (%)	dSUVmax	Median	74.68	58.14	<0.001 *
		IQR	64.25–84.25	36.07–74.20	
	dSUVmean	Median	70.17	50.79	<0.001 *
		IQR	54.34–78.57	31.58–66.28	
	dMTV (cm^3^)	Median	68.63	48.18	0.003 *
		IQR	42.81–82.49	14.76–71.75	
	dTLG	Median	89.52	73.68	<0.001 *
		IQR	79.40–95.47	50.80–88.83	

pCR, pathologic complete response; PET, positron emission tomography; SUV, standard uptake value; MTV, metabolic tumor volume; TLG, total lesion glycolysis; IQR, interquartile range; *, *p* < 0.05.

**Table 4 cancers-14-01987-t004:** Comparisons of predictive performance from conventional PET parameters, from physicians and from the ML model.

	Cutoff	AUC	Sen (%)	Spe (%)	PPV (%)	NPV (%)	ACC (%)
PET1-SUVmax	>13.15	0.592	57.4	61.7	17.7	90.9	61.2
PET1-SUVmean	>4.70	0.588	79.6	39.1	15.8	93.0	44.2
PET1-MTV (cm^3^)	>41.11	0.627	53.7	70.2	20.6	91.3	68.1
PET1-TLG	>142.97	0.635	68.5	57.1	18.9	92.7	59.1
PET2-SUVmax	≤3.97	0.687	74.1	58.8	20.5	94.0	60.7
PET2-SUVmean	≤1.83	0.726	66.7	71.5	25.2	93.7	70.9
dSUVmax	>56.5%	0.737	88.9	48.7	19.9	96.8	53.7
dSUVmean	>43.9%	0.745	94.4	42.8	19.2	98.2	49.3
dMTV (cm^3^)	>55.4%	0.625	68.5	56.6	18.5	92.6	58.1
dTLG	>86.2%	0.703	68.5	69.1	24.2	93.9	69.1
Physicians			33.9	86.4	29.2	90.8	80.5
ML model		0.977	94.4	92.2	93.7	93.1	93.4

AUC, area under curve; Sen, sensitivity; Spe, specificity; PPV, positive predictive value; NPV, negative predictive value; ACC, accuracy.

## Data Availability

Restrictions apply to the availability of these data. Data were obtained from the Samsung Medical Center and are available from the corresponding author with the permission of the Samsung Medical Center.

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
