# Peer review of "Predictive Value of 18F-FDG PET/CT Using Machine Learning for Pathological Response to Neoadjuvant Concurrent Chemoradiotherapy in Patients with Stage III Non-Small Cell Lung Cancer"

_cancers, 2022, doi:10.3390/cancers14081987_

Round 1

Reviewer 1 Report

This review is impartial and based solely on my personal opinion. I do not hold any conflict of interest that could prejudice the outcome of this report.

The article is original and addresses some very recent problems that have still not been thoroughly analyzed, although the attention paid to them is growing. To the best of my knowledge, there already are some articles that analyze radiomics in breast cancer, but just very few texts deal with the same issue when it comes to the NSCLC. At the same time, I would suggest making the title of the article more concise since the current one seems to be too long and complicated.

I anticipate more studies in this field that would entail larger numbers of patients as this could make conclusions more statistically significant and valid. To my opinion, a quite modest sample of patients is the main drawback of this article. Moreover, I see another limitation to this work: it was a retrospective study in which patients might had undergone different treatment (e.g. not the same radiation modality - 3D, IMRT or VMAT).

The application of radiomics with machine learning using 18F-fluorodeoxyglucose (FDG)-PET in patients with NSCLC is a modern and under-explored area. I still have some doubts about the misinterpretation of images after chemoRT, especially the interpretation of radiation induced pneumonitis. Overall, the results are interpreted appropriately and statistical analysis is made carefully.

The article is written in an appropriate way. All the data is presented in a comprehensive manner. Patient characteristics, diagnostic assessment and statistical analysis as well as ML model are described with sufficient details to allow another researcher to reproduce the results.

Furthermore, the section “Discussion” does take into account the strengths and weaknesses of this study.  I definitely agree with the author's view of limitations of this study.

By the way, I have also noticed one minor issue in the “References” section. To my opinion, since the main focus of the article is the NSCLC, the Reference 2 which mentions SCLC is less relevant that the rest of the references. I think we should stick to the NSCLS throughout the text and assume that Reference 2 should be corrected. Overall I find the references to be correct and the English language – appropriate and understandable.

I conclude that the paper is in principle acceptable after minor revisions are made.

Author Response

Dear editor of Cancers

We greatly appreciate the review of the original article and the helpful suggestions. Please, find below our point-by-point response to the reviewers’ comments, and a decision of the changes made to the manuscript. Revisions made in response to reviewer comments are shown in red in the revised manuscript.

Sincerely,

Joon Young Choi, M.D., Ph.D.

Department of Nuclear Medicine, Samsung Medical Center, Sungkyunkwan University School of Medicine, Seoul, Korea

Responses to Reviewer #1

This review is impartial and based solely on my personal opinion. I do not hold any conflict of interest that could prejudice the outcome of this report.

The article is original and addresses some very recent problems that have still not been thoroughly analyzed, although the attention paid to them is growing. To the best of my knowledge, there already are some articles that analyze radiomics in breast cancer, but just very few texts deal with the same issue when it comes to the NSCLC. At the same time, I would suggest making the title of the article more concise since the current one seems to be too long and complicated.

  • Thank you for your comment. As you pointed out, I have revised new title of this article as "Predictive value of 18F-FDG PET/CT using machine learning for pathological response to neoadjuvant concurrent chemoradiotherapy in patients with stage III non-small cell lung cancer".

I anticipate more studies in this field that would entail larger numbers of patients as this could make conclusions more statistically significant and valid. To my opinion, a quite modest sample of patients is the main drawback of this article. Moreover, I see another limitation to this work: it was a retrospective study in which patients might had undergone different treatment (e.g. not the same radiation modality - 3D, IMRT or VMAT).

  • I understand your concern. As you said, we don’t think the number of enrolled patients is greatly enough to perform the machine learning. However, we would like to emphasize that there has not been a research that analyzed this number of patients with non-small cell lung cancer. I agree with your suggestion which studies for a larger number of patients are needed in the near future. Relevant line was added in the Discussion section as another limitation.
  • There may be differences in radiation modality since we analyzed patient records for a period of more than 10 years. I would describe this point in addition to the limitation. Likewise, the fact that a retrospective study was performed would be additionally described in the limitation.

The application of radiomics with machine learning using 18F-fluorodeoxyglucose (FDG)-PET in patients with NSCLC is a modern and under-explored area. I still have some doubts about the misinterpretation of images after chemoRT, especially the interpretation of radiation induced pneumonitis. Overall, the results are interpreted appropriately and statistical analysis is made carefully.

  • I understand your concern. We took a very careful approach in visual assessment and texture analysis considering this point. By careful comparisons between baseline and post-CCRT images, at least we could find the center of residual primary tumor site on post-CCRT PET/CT images. By using this center, the VOI of tumor could be automatically generated by the gradient-based segmentation method (PET Edge) of the MIM software, which resulted in more accurate tumor volumes and total glycolytic activity than constant threshold methods. Previous study by Suga M, et al. has discriminated lung cancer from radiation pneumonitis (RP) in images based on accumulated 18F-FDG. They suggested that texture features can differentiate NSCLC from RP after radiotherapy more accurately than SUV parameters such as SUVmax and MTV. The intratumoral heterogeneity of 18F-FDG uptake evaluated by texture analysis improved diagnostic ability to differentiate NSCLC from RP. Although we cannot exclude the possibility that the part of radiation pneumonitis included in the automatically-generated tumor VOI, we suggest that the presented method in our study is the most appropriate considering the current available and alleged methods. However, we agree with your comments. Relevant line was added in the Discussion section as additional limitation.

The article is written in an appropriate way. All the data is presented in a comprehensive manner. Patient characteristics, diagnostic assessment and statistical analysis as well as ML model are described with sufficient details to allow another researcher to reproduce the results.

  • Thank you for your great compliment.

Furthermore, the section “Discussion” does take into account the strengths and weaknesses of this study.  I definitely agree with the author's view of limitations of this study.

  • Thank you for your great compliment.

By the way, I have also noticed one minor issue in the “References” section. To my opinion, since the main focus of the article is the NSCLC, the Reference 2 which mentions SCLC is less relevant that the rest of the references. I think we should stick to the NSCLS throughout the text and assume that Reference 2 should be corrected. Overall I find the references to be correct and the English language – appropriate and understandable.

  • I totally agree with your opinion. I would like to replace Reference 2 relevant to non-small cell lung cancer.

I conclude that the paper is in principle acceptable after minor revisions are made.

  • Thank you very much.

Thank you for your helpful comments.

Reviewer 2 Report

The study concerns the very current and still not exhausted topic of PET radiomic analysis and the possibility of applying machine learning in everyday clinical practice. The report is well structured, material and methods properly selected and described. The size of the study group is impressive and meets the needs of radiomic analysis.
The whole work is written in a legible and accessible way, it is read with interest.
The only important comment I have to the title of the publication: "Comparison with conventional PET parameters and physicians", this sentence in the fragment "comparison with physicians" is in my opinion incomprehensible and may be misleading. Please consider using a different term, maybe "physicians visual assessment", "visual assessment", "physicians assessment".

Author Response

Dear editor of Cancers

We greatly appreciate the review of the original article and the helpful suggestions. Please, find below our point-by-point response to the reviewers’ comments, and a decision of the changes made to the manuscript. Revisions made in response to reviewer comments are shown in red in the revised manuscript.

Sincerely,

Joon Young Choi, M.D., Ph.D.

Department of Nuclear Medicine, Samsung Medical Center, Sungkyunkwan University School of Medicine, Seoul, Korea

Responses to Reviewer #2

The study concerns the very current and still not exhausted topic of PET radiomic analysis and the possibility of applying machine learning in everyday clinical practice. The report is well structured, material and methods properly selected and described. The size of the study group is impressive and meets the needs of radiomic analysis.

The whole work is written in a legible and accessible way, it is read with interest.

  • Thank you for your great compliment.

The only important comment I have to the title of the publication: "Comparison with conventional PET parameters and physicians", this sentence in the fragment "comparison with physicians" is in my opinion incomprehensible and may be misleading. Please consider using a different term, maybe "physicians visual assessment", "visual assessment", "physicians assessment".

  • I understand your comment. Another reviewer pointed out that the title was too long and complicated. I have revised new title of this article as "Predictive value of 18F-FDG PET/CT using machine learning for pathological response to neoadjuvant concurrent chemoradiotherapy in patients with stage III non-small cell lung cancer".

Thank you for your helpful comments.

Reviewer 3 Report

Manuscript needs major revision

Abstract needs to  be  more  concise  and  clear

Authors need to explain in details how machine learning improves precision PET CT scan model, as well as explain in the details machine learning algorithm

Authors need  to  explain in details how ML improve resolution of the PET CT  scan  and also to show  one figure of Lung tumor image presented by classical PET CT  scan and  another with improved view by the ML PET- CT scan with enhanced illumination resolution

Authors  need  to  explain in details how ML performs analysis  of high dimensionality radiomic feature extraction  of  the clasical  PET  CT  scan

Fig 3  Legend  needs  detailed explanation

Elaborate more  on  the chemotherapy regimens  used  and  compare their efficacy

For the paclitaxel (50 mg/m2) or docetaxel (20 mg/m2) plus either cisplatin (25 mg/m2) or carboplatin and correlate with Radiotherapy used dose of 44-45 Gy

Author Response

Dear editor of Cancers

We greatly appreciate the review of the original article and the helpful suggestions. Please, find below our point-by-point response to the reviewers’ comments, and a decision of the changes made to the manuscript. Revisions made in response to reviewer comments are shown in red in the revised manuscript.

Sincerely,

Joon Young Choi, M.D., Ph.D.

Department of Nuclear Medicine, Samsung Medical Center, Sungkyunkwan University School of Medicine, Seoul, Korea

Responses to Reviewer #3

Manuscript needs major revision

Abstract needs to be more concise and clear.

  • As you said, I modified the abstract to be more concise and clear.

Authors need to explain in details how machine learning improves precision PET/CT scan model, as well as explain in the details machine learning algorithm

  • This study investigated the complex relationship between variable texture features derived from PET/CT and pCR through iterative training using machine learning. After that, test results with these trained outcomes revealed high AUC, accuracy, F1, precision, and recall. These results did not mean that the precision of the scan model improved, but rather that it showed high predictive performance when structured data such as texture features were analyzed by machine learning.
  • As you commented, I explained in the details machine learning algorithm in "2.4. Machine learning (ML) model".

Authors need to explain in details how ML improve resolution of the PET/CT scan and also to show one figure of Lung tumor image presented by classical PET/CT scan and another with improved view by the ML PET- CT scan with enhanced illumination resolution

  • As mentioned in the previous comment, our study did not evaluate improving the resolution of PET/CT scan images through ML. As you said, in the future, we will investigate PET/CT images using artificial intelligence such as deep learning and demonstrate new research to improve image resolution. Thanks for the great idea.

Authors need to explain in details how ML performs analysis of high dimensionality radiomic feature extraction of the classical PET/CT scan

  • We did not use ML to extract the radiomics features from the PET images. The radiomics features was acquired by the tool box for image analysis called CGITA facilitated by MATLAB software by using automatically-generated the tumor VOI by a gradient-based method. This method was described in the “2.3. 18F-FDG PET/CT analysis” section. The ML was used as a method to investigate the relationship between the radiomic features obtained using this method and pCR.

Fig 3  Legend needs detailed explanation

  • I described more detail in the Fig 3 Legend.

Elaborate more on the chemotherapy regimens used and compare their efficacy

For the paclitaxel (50 mg/m2) or docetaxel (20 mg/m2) plus either cisplatin (25 mg/m2) or carboplatin and correlate with Radiotherapy used dose of 44-45 Gy

  • I understand your concern. I described the treatment regimen in more detail. I also added a previously published article as a reference for the treatment regimen.
  • The optimal multimodal approaches for N2 disease remain controversial. Although definitive concurrent chemoradiotherapy is considered the standard of care, its oncologic efficacy can be limited by the high rate of local failure. Adding surgical resection to this bimodal treatment as a neoadjuvant treatment setting has been attempted and has achieved enhanced local control and improved survival, but the main concern regarding this approach is the increased risk of postoperative mortality and morbidity.

Neoadjuvant CCRT followed by surgical resection is the preferred treatment modality at our institution and has been prospectively and consecutively performed for medically fit patients with resectable NSCLC with N2 disease. Kim HK, et al. reviewed treatment outcomes including early postoperative results and long-term survival rates in patients who underwent surgical resection after neoadjuvant CCRT for NSCLC with N2 disease. Neoadjuvant CCRT followed by surgery could be performed with acceptable early postoperative outcomes, satisfactory local control, and encouraging long-term survival.

All chemotherapy administered to the patients included in this study was in accordance with the standard treatment guidelines based on the NCCN practice guideline. In our study, the main purpose of our study was the predictive value of the treatment response through machine learning rather than comparing the efficacy of chemotherapy regimen. Considering the study design and data, it is difficult to evaluate the comparative efficacy of each treatment in our study. However, as far as we know, there are lack of previous studies showing definite advantage in therapeutic efficacy between the chemotherapy regimens used in our study.

Thank you for your helpful comments.
